# The Correlation between Psychological Characteristics and Psychomotor Abilities of Junior Handball Players

**DOI:** 10.3390/children9060767

**Published:** 2022-05-24

**Authors:** Vlad-Alexandru Muntianu, Beatrice-Aurelia Abalașei, Florin Nichifor, Iulian-Marius Dumitru

**Affiliations:** Faculty of Physical Education and Sports, University „Alexandru Ioan Cuza”, 700506 Iași, Romania; beatrice.abalasei@uaic.ro (B.-A.A.); florin.nichifor@uaic.ro (F.N.); iulian.dumitru@uaic.ro (I.-M.D.)

**Keywords:** handball, motor capacity, psychological characteristics, junior players, software creation

## Abstract

The general development of the sports world has guided researchers in sports science to study excellence in sports performance, namely, the study of the characteristics and requirements specific to each sport. However, in order to meet these requirements, each individual must have a set of specific characteristics similar to those of the group to which he/she belongs. The variables in the study are related to the psychomotor abilities and psychological aspects that could influence the overall performance of junior III handball players. The main work instruments are related to field testing and psychological characteristics measurement. For psychomotor abilities, we used means such as the TReactionCo software (eye–hand coordination), Just Jump platform (dynamic balance), Tractronix system (general dynamic coordination), and Illinois test (spatial-temporal orientation), and for the psychological characteristics, we used the Motivational Persistence Questionnaire. In addition, the result of the study is represented by new software that we created in order to better observe the level of development of these characteristics in junior handball players. From a statistical point of view, we calculated the correlations between psychomotor abilities and psychological characteristics using ANOVA in order to see field position differences and performed linear regression for the variables of this study.

## 1. Introduction

The handball game, due to its complexity, requires and is equally part of the improvement of the motor manifestation mode during matches from the perspective of two components.

Players must be mentally prepared to cope with psychological discomfort resulting from the stressful effects of competitions, training periods in isolated areas, the monotony of training, affected interpersonal relationships, or even conflict situations.

Sport psychology, as a branch of psychology applied in our field, has the objective of studying the adaptations of people’s mental processes, needs for competitive activity, and training periods.

All stages of sports training, especially in the current period, contain an increased level of difficulty in the context of interdisciplinary and psychological contributions [1,2].

An individual or collective investigation from a psychological point of view can establish the level at which the athlete is able to fulfill or demonstrate as effectively as possible the skills, knowledge, and degree of training, all of which are related to the level of competition.

The psychology of sports activities, as a precisely directed scientific discipline, addresses and studies the educational and intellectual processes of athletes engaged in sports activities, with behavior and motor reaction being precisely pursued goals.

In order to obtain the best possible performance, motor activity presents physical and mental demands, and the maximum effort alternates between these two spectra of the performance of players.

The main purpose of this work is to highlight correlations between the psychological characteristics and psychomotor abilities of junior handball players. The main mental aspects that we chose were represented by motivational persistence, self-discipline, and planning capacity.

Firstly, we chose to assess motivational persistence because it can be an important factor that can help players to continue their activity at a higher level of performance, even if we are talking about junior players. Motivation mainly concerns behaviors focused on fulfilling personal goals such as joy, curiosity, satisfaction, and interest. All of these are integrated with the inner self and correspond to the value system of the athlete [3].

Secondly, self-discipline can be an important aspect in team sports and is characterized by the capacity of players to assess the situation that they face daily according to the necessities of the game.

This psychological characteristic has a much greater influence than the close follow-up of training, level of training, and skill development because a handball player must have mental self-discipline in order to make the best decisions in unpredictable environments where competitiveness is extremely high [4].

Thirdly, planning capacity has great importance in playing team handball because of the influence that it can have in daily situations that a team and the players experience. According to sports psychologists such as [5], planning should be conducted in accordance with physical training. First of all, together with the athlete and coaches, sports psychologists study the main objectives of each training cycle in order to be able to fulfill a series of mental objectives that are established.

The game of handball, as a performance sport, requires intense effort from players due to the extremely difficult demands of training and competitions that require maximum concentration, in addition to all of the physical, volitional, and mental abilities.

Sports performance, like human performance in general, is determined by logical, emotional, and creative activities of the spirit of athletes and plays a key role in obtaining the best indices of motor manifestation [6].

The main goal of high performance in sports activities is considered the borderline activity of human possibilities and multilateral mental and physical processes that evolve simultaneously with the athlete.

In recent years, the study of performance in sports based on anthropometry has shown [7]:How morphological prototypes are important for success within and among the sports phenomenon;An increased morphological variability in some sports compared to other disciplines or sports;Athletes who own or who by specific means obtain an optimal anthropometric profile for a specific sporting event are more likely to achieve success;Morphological optimization is useful for assessing training status and talent selection at both female and male levels.

The intensity of this sport varies between walking, moderate running, short-distance sprints, lateral movements, and forward/backward movements, so an increased level of specific endurance is critically needed to maintain an optimal degree of play throughout the match [8,9].

We can say that, along with body mass, other anthropometric characteristics have been shown to be crucial for sports activities and thus achieving performance in the game of handball [10], as well as in other team sports such as volleyball, football, and rugby [8]. For example, throwing can be seen as the most important technical action that players perform [11,12,13] and is dependent on the arm’s ability to achieve a sufficient degree of acceleration so that the ball leaves the hand at the maximum possible speed.

Based on the anthropometric measurements of handball players presenting different levels of practicing this sport, it was concluded that, among the players analyzed, those who showed a higher level of performance were the tallest with a large wingspan and an optimal body mass with low proportions of fat [14].

The analysis of the entire kinematic chain at the time of throwing is seen as an impractical means of evaluation, with joint mobility being a common method of quantification using traditional goniometric measurements of the area of movement. Few studies have deepened these types of measurements and their influence on mobility, and statistically insignificant results can be found in the reports of specialized studies [7,15].

Skills such as obtaining visual information about a moving object (ball and opponents) and a high degree of oculomotor coordination are advantages for players, as it allows them to react to external stimuli more effectively by adapting their whole body according to the situation in the field [14]. Other studies conducted showed that perceptual skills combined with a good ability to anticipate movements and distribute attention can lead to success in this team sport [16].

Competitions in sports games take place in environments that differ in space and time, aspects that condition the wave of information provided by opponents, and a good decision-making ability allows athletes to move on the playing surface in the best circumstances.

Authors have stated the importance of the reaction time arising from oculomotor coordination for obtaining victories [17,18,19]. The evaluation of the reaction time of athletes revealed that the teams that obtained positive results had a much higher index of reaction time, which led to winning matches.

New training methods are focused on good physical training, which is achieved by using methods and means similar to game situations and from the perspective of the psychological aspects required by the modern version of this sport [20].

Experts in this sport and coaches analyze the efficiency of the players that they train through standardized methods for each competition, so they can accurately determine the contribution of each player to the success of the team, as well as the team in general [21].

The explosive force on the lower limbs is responsible for performing various variations of jumping, speed running, and acyclic movements in the handball game. In this sport, a player must generate forces with different variations in intensity (throwing or fighting for positions on the field) [22]. This is why a differentiated load in each training session must be included for the development of maximum strength. Vertical jumps are predominant in performing most technical and tactical tasks in handball. Impulse jumps on both legs are much more common in the defensive phases (blocking the balls thrown by the opponent).

The application of different methods and means for improving the explosive force had a positive result in the case of juniors but also for those who practice this sport at a high level. The most common method for improving vertical jumping is plyometric training [1,23], followed by neuromuscular and proprioceptive training.

If we are to analyze the awareness of sports specialists, we can notice that it has increased with regard to the role of personality in developing sports performance. In competitive sports, such as handball, the players and trainers are on the lookout for better performance; therefore, great efforts are made in order to enhance the player’s achievements. While trying to obtain better performance, the actual performance of the athletes is constantly evolving alongside their psychological characteristics [24].

Further research has focused on the diversity of psychological profiles and psychological characteristics such as motivation, mechanisms involved in coping with stress, and the adaptation skills of different athletes by age, gender, and sport type [6,25,26,27,28].

In our domain (sports science), a nomothetic approach is more frequently used. Domain specialists that have the same view focus on the socio-demographics and sports characteristics of athletes and carry out comparisons by age, gender and sport type [29,30,31,32].

Alongside the assessment of some basic somatic parameters [12,33,34], motor skills [35,36], and sports seniority [36,37], it seems imperative to realize evaluations of psychomotor abilities.

Aspects such as the prediction of the adversary’s movements with or without the ball, attention, choosing the appropriate response, perception, and high levels of sensory and motor fitness are some elements that can help a team win the game [19,38]. With the aid of these psychomotor abilities and by gathering visual information through a high level of eye–hand coordination, players can react to external stimuli with more efficiency and adapt their movements according to situations that happen on the court [39].

As work objectives, we identified the most important ones, such as:Assessing the level of development of certain psychomotor characteristics;Identifying the most important psychological characteristics that could influence the activity of junior handball players;Creating a performance profile of junior III handball players;Creating a software program that could help the training process and the selection phase by observing the general development of the players.

Firstly, we assumed that the most important psychological characteristics for junior handball players are motivational persistence, planning capacity, self-discipline, implementation, and recurrence.

Secondly, we assumed, through the second hypothesis, that some psychomotor characteristics of the subjects have correlations with psychological aspects.

Lastly, the final hypothesis was to see if there is any correlation between the psychological aspects of these junior handball players, which was tested with multivariate tests.

## 2. Materials and Methods

### 2.1. Sample

For this study, we chose to apply our evaluation protocol on 181 junior III handball players from 10 teams in Romania. The research group was an average of 13.5 ± 0.5 years, and they came from different living areas (rural/urban). All of the findings were based on the results that we gathered from tests and measurements, and the analysis was conducted using statistical software.

### 2.2. Design

For the research design, we chose a set of field tests in order to assess specific psychomotor components of junior III handball players, further applying a questionnaire for some psychological characteristics such as motivational persistence, self-discipline, and planning.

The main psychomotor characteristics that we focused on were eye–hand coordination, dynamic balance, spatial orientation, and general dynamic coordination, and for each of these elements, we applied a field test with specific elements from the handball game.

All of the field tests had the main objective of testing the hypotheses of the present research study.

### 2.3. Instruments

As instruments, we used TReactionCo Sofware, laptops, special keyboards, tripods, Just Jump platform, infrared sensors, cones, and the Motivational Persistence Questionnaire for the psychological aspects; all of their actual usage in the research is presented in the Procedure subsection.

### 2.4. Data Analysis

The data presented are based on correlations, means and standard deviations, 95% confidence intervals, *p* values, Durbin Watson statistic, R-value, ANOVA analysis, and also Cronbach’s alpha.

After obtaining data from the players, all of the information was input into the statistical analysis software SPSS v. 26.

### 2.5. Procedure

Before the beginning of field testing, all players were informed about the structure of the evaluation, and the tests were explained and demonstrated so that they would have an overall view of the process. The evaluation started with the questionnaire to measure their levels of motivational persistence, self-discipline, and planning capacity. The questionnaire used is a validated one, and we also computed Cronbach’s alpha in SPSS to obtain a score reliability coefficient.

Eye–hand coordination: For the system setting, the upper limb option is selected. The athlete sits on a chair and positions the keyboard resting on his/her thighs. The laptop is placed in front of the subject at a distance allowing better observation of markings that appear on the screen. The evaluation begins when the athlete presses one of the three keys. A red dot appears on the left or right side of the screen, and the athlete has the task of pressing the key on the side where that mark appears as soon as possible after the appearance of the stimulus. Each subject has to analyze 20 successive images, and the average reaction time at the level of the upper limbs is recorded. It is considered a correct assessment if the athlete has more than 10 correct hits/touches. The average time should be as short as possible.

Dynamic balance: Depending on the type of jump that needs to be performed, the player positions him/herself on the jumping platform, performing the movement necessary to achieve the goal in order to achieve the best values (SJ, FJ, CMJ, and 4X). Depending on the distance to be covered, the distance is calculated with the help of a roulette wheel, positioning the sides on either side of the running lane to position the gates with infrared photocells. The athlete positions him/herself at the starting line and starts running at free speed and enters the deceleration process after passing the last area with the whole cell with photocells.

Spatial-temporal orientation—Illinois test description: With the help of roulette, a distance of 5 m in a straight line is measured. Heads are positioned at point 0, and 2.5 m and 5 m distances are measured. From the level of the head from a distance of 2.5 m, 3 successive distances of 3.3 m are measured with the help of the roulette wheel, at the level of which milestones are placed. At the start and finish points, two gates with infrared sensors are positioned, which start automatically after the athlete leaves, and his/her final time is recorded. The subject has to complete the route as shown below.

General dynamic coordination: Distances of 5/10/15/20 m are measured on a straight line, and the markings for the positioning of the tripods with photocells are positioned on either side of the corridor on which the athlete will run. At the beginning of the test, the athlete must run at a high tempo without major deviations in the direction of travel in order to avoid inconsistencies in the values obtained.

Agility—505 test description: The athlete must run to the 15 m marker in order to accumulate sufficient speed to move; after crossing the 5 m mark and crossing the imaginary drawn line, the athlete must run back the same distance of 5 m. The recorded time is the time in which the athlete travels a distance of 5 m (round trip). The twisting ability of each leg is tested, and the subjects are instructed that they should not exceed the 5 m line by much so as not to waste too much time.

Choosing the psychological variables in the present study was preceded by analyzing research papers, in which we tried to identify the elements that might have implications in sports practice. Generally, we identified some common aspects, such as motivation, and we considered that even at a young age, the capacity to maintain a high level of motivation can represent a true advantage in the future careers of these junior handball players. This action was followed by identifying other characteristics that we considered to be important (self-discipline and planning capacity); these two have the ability to improve the general level of the athletes. It is necessary, at any age, when practicing team sports, to have the capacity to self-organize for the tasks that are presented in training sessions, in official competitions, and during free time between these events. Continuous growth and the maintenance of a linear and constant evolution can be influenced by such factors. In addition, the capacity to plan, even for a junior player, is important, as it means that if the objectives and the purposes of the physical activities are clearly stated, the sports life of the individual can improve.

Regarding the psychological characteristics, they were evaluated by applying the Motivational Persistence Questionnaire, which contains 30 items divided into 5 categories (with 5 questions for each dimension), with a 5-step answer scale (1—to a very small extent; 5—to a very large extent) and can determine the levels of the following characteristics: motivational persistence, planning, self-discipline, determination, recurrence, and ambition [40].

-Model questions:-For motivational persistence: “I maintain my motivation even in activities that lasts for months”;-For planning: “I plan in detail what I have to do for the next day”;-For self-discipline: “Even if it’s not necessary, I put my things in order.”

## 3. Results

In the first phase, we computed reliability statistics in order to obtain the Cronbach’s alpha value for the Motivational Persistence Questionnaire that we applied in our research group. As is shown in Table 1, the 622 value puts the motivational persistence, self-discipline and planning capacity in the upper range of confidence for the specific items.

In the table below (Table 2) are presented the means of the psychological aspects after the evaluation of players alongside the number of players and standard deviation. In addition, it presents the overall value of Cronbach’s alpha for the questionnaire applied and the individual values for the items attributed to the motivational persistence, planning capacity, and self-discipline.

We also calculated the general evaluation results for the psychological aspects as well as the mean and standard deviation of the group. If we analyze the levels presented in the table above, we can see that the whole group can be qualified in the upper limit of development for the three psychological characteristics measured (Table 2).

Furthermore, we performed an ANOVA analysis (Table 3) in order to observe the main differences between the means of specific field positions for the psychological characteristics. As is shown in the table above, we obtained multiple results that are statistically significant, which led us to the conclusion that for different players that occupy a certain area in the field, the psychological characteristics are present but to different extents. This variation can be connected to the general psychological state of the junior players, and outside factors can influence their motor capacities at some point.

In the table above (Table 4), we show the Pearson correlations calculated for the field tests that we used in this study, and the significant values are highlighted. As we can observe, there is a high negative correlation between the countermovement jump and the 505 test (−0.229) and a high positive correlation between the 505 test and the 10 m run (0.418). The 10 m run also has a strong negative correlation with the countermovement jump (−0.477).

For the agility and general dynamic coordination, the r-value (0.418) signifies that there is a directly proportional influence and growth between them. This means that better indices or low results on the agility course can increase the results for the psychomotor ability mentioned. This led us to a primary conclusion that the training process of the junior handball players must be performed properly, and it must influence, in balanced proportions, all of the motor and psychomotor areas.

The dynamic balance presents a negative r-value (−0.477) with general dynamic coordination. This result means that if the dynamic balance results increase, the psychomotor ability decreases in a beneficial way by reducing the time when taking the test. The handball game, by its growth, is known as a very dynamic sport in which the players must perform multiple motor tasks, such as running and passing the ball, running–stopping–throwing, and avoiding the opponents. Thus, all of these elements can be positively influenced by the effective development of these psychomotor abilities.

Another statistical analysis that we conducted was related to the effects of the psychological aspects of the players, which were analyzed and tested through the questionnaire applied in our study (Table 5). As we can notice, there is a significance of 0.00 between motivational persistence, planning, and self-discipline, while planning capacity has a strong correlation of 0.00 with self-discipline. By gathering these results, we can state the fact that besides the motor capacities of these junior handball players, psychological aspects such as motivational persistence, self-discipline, and planning capacity are some factors that need to be taken under consideration in reference to these players and in general in the handball game.

Lastly, we performed linear regression (Table 6) in order to see which characteristics tend to have a greater influence when taken into consideration with the other variables of the study. Our output shows a relationship between psychomotor abilities (general dynamic coordination, eye–hand coordination, spatial-temporal orientation, and dynamic balance) and psychological aspects (self-discipline and motivational persistence) of 83%, which can influence the general evolution of performance when practicing this team sport.

The conclusion from these values is that in a dynamic sport that involves the continuous movement of players with and without the ball, a good level of these psychomotor abilities alongside a good psychological capacity can improve the overall output of the players; these aspects need to be taken into consideration in the training activity.

To show the influence of the variables and their weight in the overall linear regression, we highlight the standardized coefficients and the *p* values of the psychomotor and psychological aspects (Table 7). As is shown in the table, statistical significance is indicated for eye–hand coordination (*p* = 0.02), general dynamic coordination (*p* = 0.04), and self-discipline (*p* = 0.03). This led us to the conclusion that these three aspects could have a greater influence on the overall process of the evolution of junior handball players.

Another important aspect that can be highlighted relates to the software that we created in order to have a better perspective of the psychomotor and psychological characteristics of junior III handball players. The “Skills” software, by using mathematical algorithms, can create an overview of the above-mentioned aspects, and it can find utility both in the training process and in talent identification actions.

## 4. Discussion

Firstly, we assessed the level of development of psychomotor abilities of junior handball players by applying motor tests that mimic specific elements of the handball game, and the results were registered and further analyzed from a statistical point of view. We focused our attention upon eye–hand coordination, dynamic balance, general dynamic coordination, and spatial-temporal orientation, these being important abilities that can influence the handball game and can represent an important factor in the talent identification phase.

As we know, of great importance in this stage of their development, psychological aspects can influence their general evolution both in the sports area and in the social environment.

After applying the questionnaire, we can observe the level of development of these psychological characteristics, for which the players scored above average (on a scale from 1 to 10) in the case of motivational persistence (6.31), planning (6.39), and self-discipline (6.44), but for better performance, we believe that the average scores of the three components could be improved.

Furthermore, we tried to create a performance profile of junior handball players by selecting these psychomotor abilities alongside psychological aspects, these having the overall capacity of creating an overview for the players.

Some studies have neem related to the influence of supplements and their interaction with motor capacity [41], and others are related to the optimization of motor skills such as hand–eye coordination [42] and biomechanics [43], but we tried to first assess more psychomotor abilities that could influence the handball game in order to have a better overview of them.

In regard to talent identification, there are studies that have focused mainly on playing position and the month of birth [44], and the difference in our study is represented by the motor and psychological aspects that we tried to identify as important to talent identification. Other researchers have tried to identify factors for talent identification by interviewing coaches and studying psychological characteristics such as coordination and carefully planning the activity of the players [45]. This information can support our research by highlighting the importance of some psychological aspects regarding the identification of gifted players.

The level of performance of the players can be compared to a series of studies that have connected anthropometric characteristics with physical components such as agility and the ability to jump in the identification of talented players [46]. Other studies from a series of disciplines have also included some of the performance characteristics mentioned previously, these being considered important in the general assessment of junior players in order to identify talented athletes [47,48,49,50,51].

After computing linear regression results, we obtained a good relationship between psychomotor abilities and psychological aspects, and in terms of the statistical value, it was shown that the R-value (0.83/83%) puts these two aspects as general elements that can condition the level of the other variables presented in this study.

A practical application of the results of this study is the creation of the “Skills” software through which we tried to aid the training, selection, and talent identification processes. In a general overview (as can be seen in the figures below), the pages of the software are related firstly to the playing positions on the handball field, and after choosing players from the uploaded database, the evaluator or coach can choose any player that they want in order to assess their general level of psychomotor manifestation (Figure 1). There are more important aspects that can be seen from this besides psychomotor abilities; as is shown in Figure 2, there are also some somatic measurements that we consider to be important in the handball game. Lastly, the psychological aspects of the players are presented (Figure 3), all of which make it possible to observe and make the best decision for the team in terms of training, selection, and even talent identification. The software was created for that purpose using IT technologies (Python), and other studies that involve its usage are set to be written.

Some limitations of the study could be represented by the number of players analyzed; a much larger study could be conducted and might include a greater number of athletes. In addition, another limitation is related to the specialty of the literature in the sense that we could not find more research articles to compare our results with.

## 5. Conclusions

From all of the data obtained from the statistical analysis, we can highlight the need to make this kind of comparison between psychomotor and psychological factors in junior handball players. The presence of significant correlations supports the veracity of the control tests used in outlining the main characteristics of these junior players.

On the other hand, the results obtained can be seen as a starting point for further leading the training process of these junior handball players by concentrating on and leading the activity towards improving the elements presented in the study.

## Figures and Tables

**Figure 1 children-09-00767-f001:**
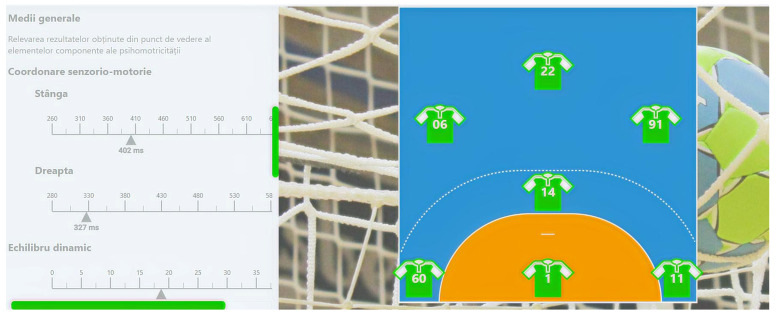
Representation of the created software’s first page.

**Figure 2 children-09-00767-f002:**
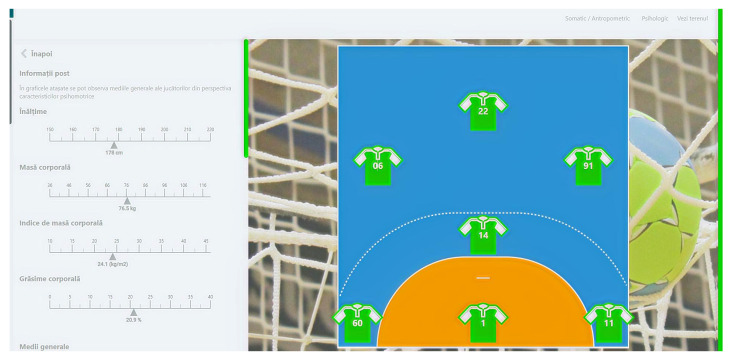
Representation of the software’s second page.

**Figure 3 children-09-00767-f003:**
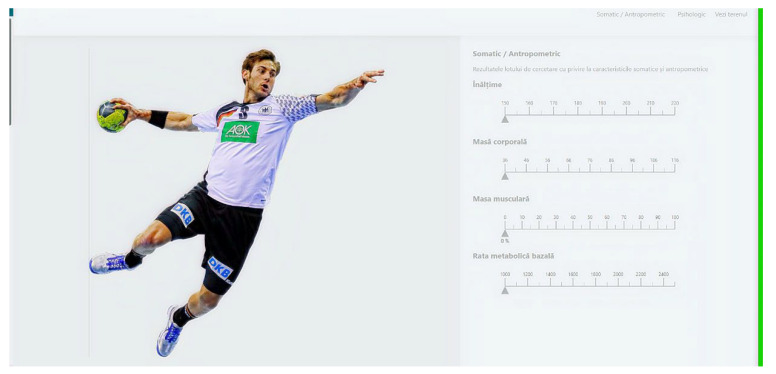
Representation of the software’s third page.

**Table 1 children-09-00767-t001:** Reliability statistics of the questionnaire applied.

Item 1	0.617
Item 2	0.606
Item 3	0.608
Item 4	0.594
Item 5	0.626
Item 6	0.590
Item 7	0.598
Item 8	0.629
Item 9	0.596
Item 10	0.609
Item 11	0.619
Item 12	0.625
Item 13	0.630
Item 14	0.624
Item 15	0.615
Cronbach’s Alpha	0.622

**Table 2 children-09-00767-t002:** Mean and standard deviation of the general scores of psychological characteristics.

	Motivational Persistence	Planning	Self-Discipline
Mean	6.31	6.39	6.44
N	181	181	181
SD	2.52	2.27	2.30

N—number of subjects; SD—standard deviation.

**Table 3 children-09-00767-t003:** Post hoc analysis. Statistically significant means of the psychological characteristics.

Dependent Variable			Sig.
Motivational persistence	Right wing	Right back	0.764
		Center back	0.003
		Left wing	0.049
		Goalkeeper	0.001
	Right back	Right wing	0.764
		Center back	0.008
		Goalkeeper	0.000
	Center back	Right wing	0.003
		Right back	0.008
		Left back	0.004
		Pivot	0.000
		Goalkeeper	0.000
	Left back	Right wing	0.931
		Center back	0.004
		Goalkeeper	0.001
	Left wing	Right wing	0.049
		Pivot	0.003
		Goalkeeper	0.000
		Left wing	0.000
		Pivot	0.031
Planning	Right wing	Right back	0.777
		Center back	0.005
		Left wing	0.020
		Goalkeeper	0.003
	Right back	Right wing	0.777
		Center back	0.013
		Left wing	0.043
		Goalkeeper	0.001
	Center back	Right wing	0.005
		Right back	0.013
		Left back	0.007
		Pivot	0.015
		Goalkeeper	0.000
	Left back	Right wing	0.936
		Center back	0.007
		Left wing	0.027
		Goalkeeper	0.002
	Left wing	Right wing	0.020
		Right back	0.043
		Left back	0.027
		Pivot	0.046
		Goalkeeper	0.000
Self-discipline	Right wing	Right back	0.777
		Center back	0.005
		Left wing	0.000
		Goalkeeper	0.000
	Right back	Right wing	0.777
		Centre back	0.012
		Left wing	0.001
		Pivot	0.712
		Goalkeeper	0.000
	Center back	Right wing	0.005
		Right back	0.012
		Left back	0.007
		Pivot	0.007
		Goalkeeper	0.000
	Left back	Right wing	0.935

**Table 4 children-09-00767-t004:** Main correlations between psychomotor testing results.

		Agility (S)	Dynamic Balance (INCH)	General Dynamic Coordination (S)
Agility (S)	Pearson Correlation	1	−0.229 **	0.418 **
	Sig. (2-tailed)		0.002	0.000
	N	181	181	181
Dynamic Balance (INCH)	Pearson Correlation	−0.229 **	1	−0.477 **
	Sig. (2-tailed)	0.002		0.000
	N	181	181	181
	Sig. (2-tailed)	0.157	0.581	0.783
	N	181	181	181
General Dynamic Coordination (S)	Pearson Correlation	0.418 **	−0.477 **	1
	Sig. (2-tailed)	0.000	0.000	
	N	181	181	181

Legend: N—number of players. ** Correlation is significant at the 0.01 level (2-tailed).

**Table 5 children-09-00767-t005:** Multivariate tests for the psychological aspects.

	F	Sig.
Motivational persistence * planning	411.89	0.000
Motivational persistence * self-discipline	603.56	0.000
	2.85	0.001
Planning * self-discipline	4262.17	0.000

**Table 6 children-09-00767-t006:** Linear regression for the variables of the study.

Model Summary ^b^				
Model	R	R Square	Adjusted R Square	Std. Error of the Estimate
1	0.917 ^a^	0.840	0.833	0.595

^a^. Predictors: (constant), self-discipline, general dynamic coordination (s), eye–hand coordination, spatial-temporal orientation(s), dynamic balance, and motivational persistence. ^b^. Dependent variable: planning.

**Table 7 children-09-00767-t007:** Statistical significance of variables in the general regression.

	Standardized Coefficients	Sig.
	Beta	
Eye–hand coordination	−0.584	0.022
Dynamic balance	0.005	0.950
General dynamic coordination	0.899	0.045
Motivational persistence	0.091	0.653
Planning	0.021	0.904
Self-discipline	0.312	0.039

## Data Availability

Not applicable.

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
