# Peer review of "The Correlation between Psychological Characteristics and Psychomotor Abilities of Junior Handball Players"

_children, 2022, doi:10.3390/children9060767_

Round 1

Reviewer 1 Report

In the “Introduction” section, it is recommended to better integrate the purpose of the work with the explanation and description of the psychological variables to be analyzed in the text. In general, the way of expressing the findings of the study should be improved, both in its description and in the comparison with previous works for a better understanding of its purpose.
It is recommended to justify the choice of the psychological variables (motivational persistence, planning capacity, self-discipline, implementation and recurrence) under study.
The objectives of the work have not been included. It is recommended to include them before the hypotheses.
Include the name of the questionnaire administered to measure the levels of motivational persistence, self-discipline and planning capacity, as well as its Cronbach's alpha.
Better organize the discussion and conclusions. The information would be better organized if the study findings were presented in connection with the objectives and hypotheses.
In the Discussion section, information is included that is expressed in the same way in the Results section. A wording that explains the findings by way of conclusion is recommended, expressing the findings of the study, but in a different format than the results, and without including numerical data. In general, the way of expressing the findings of the study should be improved, both in its description and in the comparison with previous works.
Limitations, practical applications and future lines are missing.
Update the references and put them all according to the regulations of the journal. There are references that are in different formats.

Author Response

Dear reviewer, please see the attachment.

Thank you for all the suggestions!

Reviewer 2 Report

Dear authors, I salute your efforts in producing this manuscript.

The theme is quite interesting and relevant in terms of talent selection.

I will make some comments that may help to improve the quality or clarity of the manuscript.

In the abstract, the authors should better clarify the reader regarding the instruments and variables of analysis in their study. They should also briefly inform about the statistical treatment adopted.

The authors' intention is perceptible and valid. However, the authors must contribute in the article which statistical options they used and why, in order to be evident the relationship with the objectives of the study. There are missing data from the psychological variables instrument, namely:

Name of the questionnaire, authors and publication reference of the validation of the questionnaire for Romanian, CFA fit indices of the questionnaire, how many items it has, how many items for each dimension evaluated, they must also put a model question for each dimension evaluated, which scale the questionnaire uses, how many levels does the scale have?

The results must demonstrate the reliability of psychological variables with at least Cronbach's alpha, or McDonald's omega. They must also present a study of data normality, as it is not understood how they use parametric tests of comparison and correlation.

An aspect that is not understood, how to compare the psychological variables with the physical variables with a t-test. Are they on the same scale? Or did the authors standardize the values ​​of the scales? What do they actually intend when comparing some variables with others?

The captions of the tables must be clear as to what they present, so that the reader has no doubts.

They must use a period instead of a comma in the decimal numbers shown in the tables and in the descriptive text.

Wouldn't it be more logical to test the predictive value of some variables over others, as they did in the linear regression presented?

What are the criteria for just presenting a linear regression between two specific variables? Why do they despise the predictive relationships of the others?

The discussion of the results must be done by contrasting the results against the theoretical references. It must be reworked.

The conclusions must clearly respond to the study objectives, point out limitations of the study, indicate future lines of investigation and evidence practical applications derived from the results obtained in the study.

Of the 44 publications with 5 years or less, 4 references were registered. With 10 years or less and more than 5 years we registered 17 references. The bibliography must be standardized according to the style required by the journal, as there are bibliographic references in more than one style.

Best wishes for a good work.

Author Response

(The authors gave the same response as above.)

Round 2

Reviewer 1 Report

On the one hand, the authors have carried out the recommendation in the "Introduction" section, integrating in the text the purpose of the work with the explanation and description of the psychological variables to be analyzed, as well as the choice of the same.
The objectives of the work before the hypotheses, the name of the administered questionnaire, Cronbach's alpha, limitations and practical applications have also been included.
However, in the discussion, the authors have to write the findings of the study including other works that support or not the results obtained in order to contrast said findings with other previous works.

Author Response

Dear reviewer, please see the attachment. Thank you for all your patience and suggestions!

Reviewer 2 Report

Dear authors, I praise your effort to improve the manuscript.

Your work has improved a lot in the procedural description.

It is recommended, however, that the following elements can be improved.

Table 1 does not justify being isolated. Cronbach's alpha value can and should be included in table 2. The alpha value is required for each of the psychological variables and not just the general algae. They must include this data in table 2.

In linear regression, we found that the independent variables explain the dependent variable in high percentage. However, what weight does each dependent variable have on the independent variable? If you want to trace a software, these data are very important to be known.

In the review, it is necessary to improve the contrast between the results obtained and indicators evidenced by the literature. Look for other talent selection jobs to make this contrast.

The conclusion should give a clear answer to each of your goals, which it does not.

Best wishes for a good work.

Author Response

Dear reviewer, please see the attachment. Thank you for all your patience and suggestions!

This manuscript is a resubmission of an earlier submission. The following is a list of the peer review reports and author responses from that submission.

Round 1

Reviewer 1 Report

INTRODUCTION:

The theoretical revision is appropriate with research topic. The introduction section is very broad and very well structured. The authors make an extensive tour of recent scholarship.

However, it is not justified the relevance/impact of the selected topic.

There is no research objectives and hypothesis established. I consider it essential in a research to establish objectives and hypothesis.

MATHERIAL AND METHODS:

The material and methods section is not correct.

The research hypotheses should not be in this section, but at the end of the introduction.

In turn, it must contain subsections: sample, design, instruments, data analysis, and procedure.

RESULTS:

The results section is not correct.

The format of the tables is wrong, there is no information as note of the table, and results have been indicated in yellow and red. The statistical analyzes are very basic.

CITES/REFERENCES:  

The numbering of citations in the text is not correct. The first one (page 1, line 37) is number 5, when it should be number 1.

The authors must review the list of references, the numbers are repeated from the first reference.

Reviewer 2 Report

I would first like to congratulate authors for conducting this study and acquiring quite large sample of young handball players. However, there are several important flaws of the submitted manuscript and potentially the study as whole. First of all, as this is a study conducted on children you must state that there was parental approval provided (you have only stated that is was approved by ethical board). The paper itself is strange and has too long introduction without clear hypotheses stated at the end. The methods are written in laymen terms which is not OK. On one occasions you are stating that psychological questionnaire was validated by SPSS software which of course is not possible. If non-validated questionnaire was used, then you first need to validated before use it on an independent sample (as we don't know if it measures what it is supposed to measure). Additionally, in tables there are no units. Discussion is very short, especially compared to introduction and conclusion again is unusually long, where authors express their point of view which is not always supported by the findings. My advice is to reject, rewrite and resubmit, but before that authors should go through some of the MDPI published papers and notice the differences with this submission.